# Microalga *Nannochloropsis gaditana* as a Sustainable Source of Bioactive Peptides: A Proteomic and In Silico Approach

**DOI:** 10.3390/foods14020252

**Published:** 2025-01-14

**Authors:** Samuel Paterson, Laura Alonso-Pintre, Esperanza Morato-López, Sandra González de la Fuente, Pilar Gómez-Cortés, Blanca Hernández-Ledesma

**Affiliations:** 1Department of Bioactivity and Food Analysis, Institute of Food Science Research (CIAL, CSIC-UAM, CEI UAM+CSIC), Nicolás Cabrera 9, 28049 Madrid, Spain; samuel.paterson@csic.es (S.P.); laura.alonsopintre@gmail.com (L.A.-P.); 2Proteomics Core Facility, Centro de Biología Molecular Severo Ochoa (CBM), CSIC-Universidad Autónoma de Madrid, Nicolás Cabrera 1, 28049 Madrid, Spain; emorato@cbm.csic.es; 3Biocomputational Core Facility, Centro de Biología Molecular Severo Ochoa (CBM), CSIC-Universidad Autónoma de Madrid, Nicolás Cabrera 1, 28049 Madrid, Spain; sandra.g@cbm.csic.es

**Keywords:** microalgae, proteomics, functional analysis, in silico digestion, *Nannochloropsis* sp.

## Abstract

The impact of the world’s growing population on food systems and the role of dietary patterns in the management of non-communicable diseases underscore the need to explore sustainable and dietary protein sources. Although microalgae have stood out as alternative sources of proteins and bioactive peptides, some species such as *Nannochloropsis gaditana* remain unexplored. This study aimed to characterize *N. gaditana*’s proteome and evaluate its potential as a source of bioactive peptides by using an in silico approach. A total of 1955 proteins were identified and classified into functional groups of cellular components, molecular functions, and biological processes. In silico gastrointestinal digestion of identified proteins demonstrated that 202 hydrophobic and low-molecular-size peptides with potential bioactivity were released. Among them, 27 exhibited theorical antioxidant, antihypertensive, antidiabetic, anti-inflammatory, and/or antimicrobial activities. Seven of twenty-seven peptides showed ≥20% intestinal absorption, suggesting potential systemic effects, while the rest could act at local level. Molecular docking demonstrated strong affinities with key enzymes such as MPO, ACE, and DPPIV. Resistance to the digestion, capacity to be absorbed, and multifunctionality were demonstrated for peptide FIPGL. This study highlights *N. gaditana*’s potential as a sustainable source of novel potential bioactive peptides with promising local and systemic biological effects.

## 1. Introduction

In recent years, non-communicable diseases (NCDs) have been considered a serious disease burden recognized worldwide. In fact, with more than 50% of all deaths being attributable to them, NCDs such as stroke, cancer, ischemic heart disease, chronic kidney disease, and auto-immune and neurodegenerative conditions are now some of the biggest threats and challenges to human health [1]. The main risk factors of NCDs can be classified into the categories of self-management, genetics, medical conditions, and socio-demographic and environmental factors, including diet [2]. Therefore, the role of dietary patterns and evidence-based nutrition interventions in the prevention and management of NCDs should be a global health priority.

Predictive models of the global population have raised concerns about the sustainability of food systems. Among the myriad strategies, a gradual transition from meat-based to plant-based diets is supported and coordinated by the main health and environmental institutions is considered an effective approach [3]. Proteins derived from animal sources are highly valued for their nutritional and functional qualities; however, livestock production significantly impacts the environment, highlighting the importance of balancing all dietary protein sources to meet human needs [4]. Among the different alternative protein sources currently being studied, microalgae are considered one of the most promising due to their optimal nutrition profile, minimal environmental footprint, and significant biological potential [5].

Compared to dry plant seeds, microalgae have been reported to have remarkably higher protein contents, ranging from 30% to 70% of dry weight (with an average value of nitrogen-to-protein conversion factor of 4.78) [6]. In addition, microalgae offer significantly greater protein yield per unit area compared to wheat or beef, owing to their simplified growth processes, reduced energy expenditure, and superior photosynthetic efficiency [7]. Furthermore, microalgal proteins are recognized for their excellent promising functional properties, low antigenicity [8], and high nutritional quality, providing all 20 amino acids (AAs) and varying amounts of essential AAs (EAAs) [9]. The AA composition of many microalgae proteins such as *Nannochloropsis gaditana* is similar to that of typical plant protein sources, such as soybean [10], and closely aligns with the FAO/WHO/UN reference profile [11]. Thus, microalgal proteins can be regarded as an alternative protein source with nutritional and biological attributes after being broken down in the gastrointestinal tract, liberating EAAs and potential bioactive peptides [12].

Bioactive peptides are small fragments, between 2 and 20 AA, which do not exert any effect within their origin protein but can exert multiple biological properties once released. Different bioactivities have been reported to contain bioactive peptides such as antioxidant, anti-inflammatory, antidiabetic, and immunomodulatory activities, among others [13]. In the case of microalgal proteins, several studies have mentioned their ability to produce bioactive peptides after enzymatic hydrolysis. Thus, proteins from *Chlorella vulgaris*, *Arthrospira platensis*, and *A. maxima* have been demonstrated to generate bioactive peptides with anti-inflammatory and antioxidant activities after their hydrolysis with pepsin, trypsin, or the combination of alcalase and papain [14,15,16]. Similarly, *N. oculata* has been reported to generate protein hydrolyzates and peptides with angiotensin-converting enzyme (ACE)-1-inhibitory, antithrombotic, and calcium/calmodulin-dependent kinase II (CAMKII)-inhibitory activities [17,18]. Saleh et al. reported that a protein hydrolyzate from *N. gaditana* with alcalase exerted high in vitro antioxidant activity [19].

Proteases, such as pepsin in the stomach and trypsin and chymotrypsin in the small intestine, play a crucial role in breaking down proteins into smaller fragments. In the case of microalgae, several studies have investigated the release of bioactive peptides from microalgae under simulated gastrointestinal conditions. For example, Li et al. [20] described ACE- and dipeptidyl peptidase (DPP)-IV-inhibitory peptides contained in *C. pyrenoidosa* digests. The study of Pei et al. [21] identified the antioxidant and anti-inflammatory peptide EMFGTSSET released after simulating gastrointestinal digestion of *Isochrysis zhanjiangensis*. The susceptibility of other microalga species to the action of digestive enzymes has also been demonstrated, such as for *N. oculata* [22]. However, to our knowledge, no studies have demonstrated the potential of *N. gaditana* as a source of bioactive peptides after its gastrointestinal digestion.

The study of bioactive peptides using in vivo, cellular, or animal models is expensive, time-consuming, and might also have ethical considerations. Hence, in silico bioinformatic approaches, which cannot only help to predict the formation, breakdown, and absorption of bioactive peptides from any known protein source but can also help to analyze the structure/function relationship, have gained a new relevance in this scientific field, as they can serve as a screening tool to accelerate and complement the discovery process of potentially encrypted bioactive peptides [23]. Therefore, the goal of this work was to carry out a complete characterization of the microalga *N. gaditana*’s proteome, and to demonstrate its potential as a source of bioactive peptides through an in silico gastrointestinal digestion strategy, supporting the value of this microalga as a sustainable alternative protein with both nutritional and health-promoting properties.

## 2. Materials and Methods

### 2.1. Samples and Reagents

Commercial *N. gaditana* microalgae biomass was supplied by AlgaEnergy S.A. (Madrid, Spain). Acetonitrile (ACN), trifluoroacetic acid (TFA), calcium chloride (CaCl_2_), Tris-HCl, formic acid, dithiothreitol (DTT), and iodoacetamide were purchased from Sigma-Aldrich (St. Louis, MO, USA).

### 2.2. Proteomic Identification and Characterization

#### 2.2.1. Protein Content and AA Profile

The protein concentration of *N. gaditana* biomass was measured via the bicinchoninic acid (BCA) method, which was performed using the Pierce BCA kit (Thermo Scientific, Waltham, MA, USA) [24]. The standard used was bovine serum albumin (BSA) at concentrations ranging from 50 to 1000 µg/mL. The AA content was determined in duplicate by using a Biochrom 30 series Amino Acid Analyser (Biochrom, Cambridge, UK), and following the previously described methodology [25].

#### 2.2.2. In-Gel Digestion (Stacking Gel)

*N. gaditana* biomass was suspended in 50 µL of sample buffer and then applied onto 1.2 cm-wide wells of a conventional SDS-PAGE gel (0.75 mm thick, 4% stacking, and 10% resolving; Bio-Rad, Hercules, CA, USA) following the methodology previously reported [26]. The run was stopped when the front entered 3 mm into the resolving gel, so the whole proteome became concentrated in the stacking/resolving gel interface. The unseparated protein bands were visualized by Coomassie staining (Bio-Rad), excised, cut into cubes (2 × 2 mm), and placed in 0.5 mL microcentrifuge tubes [27]. The gel pieces were distained in ACN:H_2_O (1:1, *v*:*v*), reduced and alkylated (disulfide bonds from cysteinyl residues were reduced with 10 mM DTT for 1 h at 56 °C, and then thiol groups were alkylated with 10 mM iodoacetamide for 30 min at room temperature in darkness), and digested in situ with sequencing-grade trypsin (Promega, Madison, WI, USA), as described by Shevchenko et al. [28]. The gel pieces were shrunk by removing all liquid using sufficient ACN. ACN was pipetted out, and the gel pieces were dried in a speedvac and re-swollen in 100 mM Tris-HCl pH 8, 10 mM CaCl_2_ with 60 ng/mL trypsin at a 5:1 protein/enzyme (*w*/*w*) ratio. The tubes were kept in ice for 2 h and incubated at 37 °C for 12 h. Digestion was stopped by the addition of 1% TFA. Whole supernatants were dried down and then desalted using OMIX Pipette tips C18 (Agilent Technologies, Santa Clara, CA, USA) before the analysis. Digestion was conducted in the presence of 0.2% RapiGest (Waters, Milford, MA, USA).

#### 2.2.3. Reverse-Phase Liquid Chromatography (RP-LC-MS/MS) Analysis (Dynamic Exclusion Mode)

The desalted protein digest was dried, resuspended in 10 µL of 0.1% formic acid, and analyzed by RP-LC-MS/MS in an Easy-nLC 1200 system coupled to an ion trap LTQ-Orbitrap-Velos-Pro hybrid mass spectrometer (Thermo Scientific, Waltham, MA, USA). The peptides were concentrated (on-line) by reverse-phase chromatography using a 0.1 mm × 20 mm C18 RP precolumn (Thermo Scientific) and then separated using a 0.075 mm × 250 mm bioZen 2.6 µm Peptide XB-C18 RP column (Phenomenex, Torrance, CA, USA) operating at 0.25 μL/min. Peptides were eluted using a 180 min dual gradient. The gradient profile was set as follows: 5–25% solvent B for 135 min, 25–40% solvent B for 45 min, 40–100% solvent B for 2 min, and 100% solvent B for 18 min (Solvent A: 0.1% formic acid in water; solvent B: 0.1% formic acid, 80% ACN in water). Electrospray ionization (ESI) was performed using a Nano-bore emitters Stainless Steel ID 30 μm (Proxeon, Odense, Denmark) interface at 2.1 kV spray voltage with an S-Lens of 60%. The Orbitrap resolution was set at 30,000 [29]. Peptides were detected in survey scans from 400 to 1600 amu (1 μscan), followed by twenty data-dependent MS/MS scans (Top 20), using an isolation width of 2 u (in mass-to-charge ratio units), normalized collision energy of 35%, and dynamic exclusion applied for 60 s periods. Charge-state screening was enabled to reject unassigned and singly charged protonated ions.

#### 2.2.4. Data Processing

Peptide identification from raw data was carried out using PEAKS Studio v11.5 search engine (Bioinformatics Solutions Inc., Waterloo, ON, Canada). Database search was performed against UniProt-Nannochloropsis gaditana.fasta (15,363 entries); UniProt release 11/2023. The following constraints were used for the searches: tryptic cleavage (semi specific) after arginine (R) and Lysine (K), up to two missed cleavage sites, and tolerances of 20 ppm for precursor ions and 0.6 Da for MS/MS fragment ions. The searches were performed allowing for optional methionine (M) oxidation and cysteine (C) carbamidomethylation. False discovery rates (FDRs) for peptide–spectrum matches (PSMs) and proteins were limited to 0.01. Only those proteins with at least two unique peptides being discovered from LC/MS/MS analyses were considered reliably identified [27,30,31,32].

### 2.3. Proteomic Functional Analysis

The proteomic functional analysis of the identified proteins from *N. gaditana* biomass was carried out using the functional analysis module from OmicsBox 3.3.2 bioinformatic software (Biobam, Valencia, Spain) through the BLAST2GO methodology [33]. The identified proteins were classified into three different functional groups: cellular component, molecular function, and biological process. The database used to perform the alignment was Non-redundant protein sequences (nr v5), filtered by taxonomy: *N. gaditana* (code 72520). The specific workflow followed for the functional analysis is shown in Appendix A.

### 2.4. In Silico Gastrointestinal Digestion of Microalga Proteins

The in silico gastrointestinal digestion of proteins was performed using the Rapid Peptides Generator (RPG) software v22.2.3 [34]. By using the Python (3.12) system, the complete data obtained from the proteomic identification were entered in fasta format, and firstly digested with pepsin (code 34) to simulate gastric digestion. Afterward, only peptides released from proteins that generated ≥100 peptides/protein after the gastric phase were selected to be concurrently digested with trypsin (code 42) and chymotrypsin high specificity (code 15). In all in silico gastric and gastrointestinal digests, free AAs and duplicate peptides were deleted, and the resulting peptides were classified based on their size and AA sequence.

### 2.5. Bioactivity Prediction by In Silico Analysis

For the prediction of the bioactivity, an in silico analysis of the peptides released during the gastric and gastrointestinal phases was conducted using the server based on a novel N-to-1 neural network Peptide Ranker [35]. Peptides with ≥0.8 value of the probability of being bioactive were selected to estimate their (i) antioxidant activity by using the Antioxidative peptide predictor (AnOxPP) database [36], (ii) anti-inflammatory activity using the PreAIP database [37], (iii) antimicrobial activity using the CAMPR_4_ database [38], (iv) antihypertensive activity using the AHTpin database [39], and (v) DPP-IV inhibitory activity by using the StackDPPIV database [40].

### 2.6. Physicochemical and Pharmacokinetic Analysis

The peptides with 4 or more predicted bioactivities were selected, and SwissADME molecular sketcher, based on ChemAxon’s Marvin JS 24.3.0 (re389bc4da912), 17 July 2024, was used to obtain the canonical SMILE for each peptide and to predict the physicochemical and pharmacokinetic properties such as molecular formula, weight, lipophilicity, water solubility, bioavailability score, AMES toxicity, and the percentage of human intestinal absorption through the SwissADME web tool (http://www.swissadme.ch/) [41,42].

### 2.7. Peptide Molecular Docking

The peptide molecular interactions with human angiotensin-converting enzyme (ACE), DPP-IV, and myeloperoxidase (MPO) enzymes were predicted using molecular docking. The crystal structure for ACE (2C6N), DPP-IV (1NU8), and MPO (3F9P) was obtained from a protein data bank (http://www.rcsb.org/) [43]. Peptides were designed using Marvin Sketch software (Chemaxon, version 19.22.0); flexible torsions, charges, and grid sizes were assigned using Autodock Tools; and docking calculations were performed using AutoDock Vina, where the binding pose with the lowest binding energy was selected as representative to visualize in the Discovery Studio 2016 Client (Dassault Systemes Biovia Corp R), as was previously described [44,45].

To clarify the experimental design, the workflow of the present research is summarized in Figure 1.

## 3. Results and Discussion

### 3.1. Proteome of the Microalga Nannochloropsis gaditana

The protein content of *N. gaditana* biomass was 21.69%. Values reported in the literature ranged from 19–24% for *N. oculata* [22] to 44.9% for *N. gaditana* [46]. Levasseur et al. [47] also compiled intermediate percentages for this genus. The protein content of *Nannochloropsis* sp. microalgae has been demonstrated to be affected by numerous biotic and abiotic factors and genomic variation among species [48]. These factors can also impact the AA profile and, consequently, the protein quality [49]. The AA profile of the *N. gaditana* used in the present research is shown in Table 1. All AAs were analyzed except tryptophan, as the methodology conditions degraded it. The measurement showed a high AA content, leucine being the major EAA, followed by lysine and phenylalanine. Among non-essential AAs (NEAAs), glutamic acid was the most predominant, followed by aspartic acid and proline. Comparing the results obtained with the recommendations of the FAO/WHO/United Nations University (UNU) report (2007), the requirements established for each AA were covered [50]. Recent studies have provided insights into the factors contributing to the high concentration of glutamic acid in microalgae, including *Nannochloropsis* sp. The findings suggest that glutamic acid’s predominance might be linked to its central role in nitrogen metabolism and assimilation through the glutamate–glutamine cycle; its function as a precursor of other amino acids like glutamine, proline, and arginine; its role in stress responses to environmental conditions by acting as an osmotic regulator that helps to stabilize cellular structures during such stress responses; and its role in the carbon and nitrogen balance of *Nannochloropsis* sp., which are high biomass producers that might optimize nitrogen assimilation through the accumulation of glutamic acid [51,52,53].

After the in-gel digestion and consequent RP-LC-MS/MS analysis of *N. gaditana* biomass, by using PEAKS and SPIDER search tools [54], a total of 13,097 peptide–spectrum matches (PSMs), 12,574 scans (which account for MS/MS scans that have a peptide spectrum associated with them), 6991 features identified with database search only, 6903 peptide sequences with modifications not including isoleucine/leucine differentiation, and 5820 features without modifications and isoleucine/leucine differentiation were identified.

A total of 1955 proteins were detected through PEAKS software. Since sequences constituted the key for homology-based protein identification, PEAKS works with protein groups that contain all the proteins that have been identified with the same peptides [26,55]. In our case, 1439 protein groups were identified. Of 1955 proteins, 1258 proteins were identified with two or more unique peptides. Their correspondent accession number, −10logP value, number of peptides and unique peptides, the type of modification present in the peptide (PTM), the average mass, and the description of all proteins detected in *N. gaditana* biomass were collected and made available in the Appendix A.

*N. gaditana*’s proteome was first studied by Fernández-Acero et al. [56], identifying 1950 proteins. The differences found in comparison with our results could be due to factors such as the strain and the growing conditions, which have been demonstrated to affect microalgae’s macronutrient distribution [57]. Hulatt et al. [58] identified 3423 proteins in *N. gaditana* biomass, where the abundance of 1543 and 113 of them was demonstrated to be influenced by the amount of nitrogen and phosphorous available in the culture conditions, respectively. In general, the limited count of proteins and peptides identified by the database search should be attributed to the incompleteness of the sequenced *N. gaditana* proteome and the differences in the experimental procedures. Thus, mapping the *N. gaditana* proteome may aid to understand variables affecting its proteins and to explore their properties as sources of bioactive peptides.

### 3.2. Functional Analysis of Nannochloropsis gaditana Proteome

To elucidate the functionality of the identified proteins in *N. gaditana*, an alignment of the protein sequences detected was carried out using the BLAST2GO methodology, classifying proteins with differentiated biological functions into three groups: biological process, cellular component, and molecular function. The functional distribution of the identified proteins at levels 1 and 2 of each functional group are shown in Figure 2, and the detailed complete level distribution of each function is available in Appendix A.

The distribution of proteins at level 1 was quite homogeneous, since the percentages associated with each function were similar (Figure 2A), with 1309 sequences for biological process (31.56%), 1424 sequences for molecular functions (34.44%), and 1415 sequences for cellular components (31.56%). However, within the sequences associated with cellular components at level 2 (Figure 2B), a noticeable difference was observed since the majority were found within cellular entities (74.13%) and not forming complexes (25.87%). Regarding cellular entities, intracellular anatomical structures and organelles were the most abundant groups (Appendix A). Concerning the biological process group level 2 classification (Figure 2C), protein sequences related to cellular processes were the most abundant (66.24%), in contrast with proteins related to localization, response to stimulus, and regulation processes. Regarding those cellular processes, metabolic processes in which energy is obtained from the metabolization of organic substances were the most abundant. Finally, concerning molecular function level 2 classification (Figure 2D), the major percentage of sequences were associated with binding processes (47.30%) and catalytic activities (44.32%) which are related to proteins associated with cyclic organic compounds, necessary for photosynthesis, mobilizing metabolites, and building structural components, such as the cell wall, among other previously studied functions [59]. Our results were in tune with the GO classification conducted by Fernández-Acero et al. [56] and Hulatt et al. [49] as the major percentage of proteins that belonged to the biological process group where related to biosynthetic processes, carbon fixation, ATP synthesis, and AA regulation, and the major percentage of proteins who belonged to the molecular process group were related with ion binding processes.

### 3.3. In Silico Gastrointestinal Digestion

The in silico gastrointestinal digestion of the complete set of 1955 proteins detected through PEAKS software v11.5 was conducted. This allowed the inclusion of proteins that could be underrepresented or absent due to technical limitations, such as low abundance, suboptimal extraction methods, or inefficient ionization in mass spectrometry analysis [60]. In this way, we would be ensuring that even proteins with marginal or undetected peptides were considered, potentially uncovering biologically relevant or functional insights [61]. Moreover, the additional 697 proteins with fewer than two unique peptides may still hold critical roles in metabolic pathways or cellular processes but be overlooked in traditional proteomics workflows. Therefore, by considering the entire dataset, researchers can identify proteins with potential bioactivity, nutritional value, or industrial applications that might otherwise be disregarded. Still, it should be noted that many organisms produce bioactive peptides in cellulo prior to any further gastrointestinal digestion, which sometimes are even resistant to the action of digestive enzymes [62]. Given the experimental limitations of the present work, the inherent bioactive peptides of *N. gaditana* were not identified, although their potential contribution to the bioactive effects of this microalga should not be discarded.

Consequently, an in silico gastric digestion with pepsin was initially performed, generating a total of 164,503 sequences that, after removing free AAs and duplicate peptides, resulted in 47,982 sequences. Then, the number of peptides generated from each identified protein was counted, and 24 proteins were selected as the most peptide-generating proteins (with ≥100 peptides/protein) after the gastric phase (Table 2).

The average mass of the selected proteins ranged from 118 to 552 kDa. The three most peptide-generating proteins were zinc finger ZZ-type (which generated 376 peptides), acetyl CoA carboxylase (207 peptides), and bromodomain containing 1 (169 peptides), the average masses of which were slightly higher than the rest. These three proteins have been previously described in other microalgae species such as *Schizochytrium* sp., *C. vulgaris*, and *Chlamydomonas reinhardtii*, mainly exhibiting a regulatory role in microalgae’s lipid metabolism and biosynthesis [63,64,65,66]. In addition, Lin et al. [67] specifically highlighted the involvement of acetyl CoA carboxylase in the type II fatty acid synthesis pathway, which occurs in the plastids of the cells of *N. gaditana*. These proteins would be of great importance since the microalga *N. gaditana* has aroused great interest due to its high lipid content and its potential as a sustainable source of omega-3 fatty acids.

Proteins’ susceptibility to the action of pepsin can be influenced by several structural and chemical factors, such as protein molecular weight, AA composition, and acidic conditions, among others. Pepsin is a protease that specifically cleaves peptide bonds at the amino-terminal side of hydrophobic and aromatic residues sites such as phenylalanine, tyrosine, and leucine, breaking polypeptide chains into smaller peptides [68]. Generally, larger proteins commonly provide more cleavage sites for pepsin due to their longer polypeptide chains [69]. Therefore, the susceptibly of the three proteins with most peptides generated could be explained by their high average mass (552, 235, 245 kDa, respectively) and the number of hydrophobic AAs found in *N. gaditana* biomass, such as leucine or phenylalanine. In addition, in the RPG software v22.2.3, pepsin was selected at pH 1.3 (code 33) in attempt to confirm its high cleavage affinity and most optimal conditions. Nevertheless, information regarding *N. gaditana* protein susceptibility to pepsin is still scarce and should be further explored.

The selected proteins generated a total of 3160 peptides during the gastric phase. This number was experimentally affordable and representative of the total identified peptides, although the potential contribution of other peptides should also be taken into account. Among them, 104 were predicted by the Peptide Ranker software tool (http://distilldeep.ucd.ie/PeptideRanker/) to be active with a score ≥ 0.8. These peptides were analyzed by using the AnOxPP, PreAIP, CAMPR_4_, AHTpin, and StackDPPIV databases to predict their antioxidant, anti-inflammatory, antimicrobial, antihypertensive, and antidiabetic activities, respectively. Three peptides, FHPKF, FIPGL, and FRPLPPLA, were predicted to potentially exert the five bioactivities assayed, while two sequences, PLPAGRF and HHDCF, were predicted to have four of those bioactivities. In addition, a second phase simulating intestinal digestion was conducted on 3160 peptides generated during the gastric phase, using trypsin and chymotrypsin as pancreatic enzymes. A total of 6224 peptides were generated after eliminating free AAs and duplicates. Of these peptides, 54 were also present in the in silico gastric digestion, indicating their capacity to resist the action of trypsin and chymotrypsin. After being analyzed with the Peptide Ranker software tool, 202 of 6224 peptides showed a score ≥ 0.8. The peptide profile is shown in Figure 3.

Most of the potentially bioactive peptides released after the complete gastrointestinal digestion were tripeptides (88), followed by tetrapeptides (47) and pentapeptides (39) (Figure 3A). The major AAs present were phenylalanine (F) followed by proline (P), arginine (R), and glycine (G) (Figure 3B). The high presence of phenylalanine (F) and proline (P) observed in the in silico gastrointestinal digestion agreed with the high content of these AAs present in the *N. gaditana* biomass (Table 1). It has been previously described that numerous bioactive peptides present a structure–activity relation, being a key part of their potential biological effects [70,71]. In tune with our results, in general and especially in the case of the antioxidant and ACE inhibitory activities, it has been described that active fragments are either dipeptides or tripeptides containing hydrophobic AAs, such as phenylalanine or proline [72,73]. In the case of microalgae-derived bioactive peptides, several microalgae species like *Chlorella* sp., *Navicula* sp. and *Arthorspira* sp. have been described as sources of small bioactive peptides with hydrophobic AAs [74,75], which suggests that *N. gaditana* could also have the potential to become a source of bioactive peptides. After the in silico digestion and bioactivity prediction analysis, from 202 peptides, 2 sequences (FIPGL and YLPPR) were predicted to exert five activities, while 25 sequences showed four of five activities (Table 3).

Regarding the potentially bioactive dipeptides generated, a search of their sequences in the BIOPEP-UWM database [76] showed that the dipeptide FK was previously described in sweet potato hydrolyzates and associated with ACE inhibitory and anti-hyperuricemic properties [77,78]. The dipeptide FR has also been described to display ACE and DPPIV inhibitory actions [79,80], as it has been demonstrated for the GF sequence [80,81]. ACE inhibitory effects were also observed for RF peptide [82,83]. In the case of the tripeptide ARF, it has been recently described that this specific sequence can not only inhibit ACE, but also its homologue, ACE2 [84]. Regarding the rest of the peptides, a search in the Uniprot Database reveled that FIPGL could be found in the protein “Glycoprotein G” of human herpesvirus 2 (strain HG52; UniProtKB entry P10211) and that the peptides KSPGW, YLPPR, and FLPPAL were present in the protein “Putative uncharacterized protein” from *Bacillus subtilis* (strain 168; UniProtKB entry P37595). However, no data about their bioactivity have been reported yet, which reinforces the potential of *N. gaditana* as a sustainable source of both known and new bioactive peptides. The predicted bioactivity of peptides may be influenced by the positions of AAs within their sequence. For instance, the placement of hydrophobic or branched-chain AAs at the C-terminal or the presence of aliphatic AAs at the N-terminal can enhance radical scavenging of reactive oxygen species or the ACE-inhibitory activity of the peptides [85].

**Table 3 foods-14-00252-t003:** Physicochemical and pharmacokinetic properties of peptides released from in silico gastrointestinal digestion of *Nannochloropsis gaditana* and predicted to have 4 or more bioactivities.

Peptide	Molecular Weight (Da)	Lipophilicity (MLogP) ^a^	Bioavailability Score ^b^	Water Solubility (log mol/L) ^c^	% Intestinal Absorption ^d^	AMES Toxicity ^e^
FK	293.36	0.6	0.55	−2.818	35.3	No
FR	321.37	0.15	0.55	−2.643	21.61	No
GF	222.24	0.34	0.55	−1.85	41.89	No
RF	321.37	0.14	0.55	−2.617	21.43	No
ARF ^1^	60.06	−1.6	0.55	0.824	71.496	No
DPMP	458.53	−1.11	0.11	−2.27	0	No
FHPR	555.63	−1.46	0.17	−2.875	6.011	No
FSPR	505.57	−1.56	0.17	−2.835	0	No
HPKF	527.62	−1.11	0.17	−2.815	17.04	No
MPPR	499.63	−1.01	0.17	−2784	6.78	No
VPGF	418.49	−0.1	0.55	−2.524	28.49	No
APMRP	570.71	−1.53	0.17	−2.922	0	No
FIPGL ^1,2^	545.67	0.15	0.17	−3.174	21.02	No
GPGCG	389.43	−2.73	0.55	−2.516	2.96	No
KAPPF	558.67	−0.6	0.17	−2.833	16.56	No
KSPGW1	573.64	−2.17	0.17	−2.862	0.134	No
PCMIR	618.81	−1.32	0.17	−2.889	0	No
PFGNR	589.64	−2.49	0.17	−2.889	0	No
PRPMR	655.81	−1.98	0.17	−2.889	0	No
RRCLF ^1^	693.86	−1.22	0.17	−2.894	0	No
WWGGV	603.67	−0.74	0.17	−3.015	19.76	No
YLPPR2	644.76	−0.84	0.17	−2.904	19.333	No
AVMPIF	676.87	−0.07	0.17	−3.012	11.265	No
EFPMIR	791.96	−1.31	0.17	−2.894	0	No
FARPGL ^1^	659.78	−1.51	0.17	−2.922	0	No
FGPQGG ^1^	561.59	−2.77	0.17	−2.977	0.112	No
FLPPAL ^1^	656.81	0.12	0.17	−3.29	17.79	No

^a^ Lipophilicity expressed as the partition coefficient between octanol and water [86,87,88]. ^b^ Bioavailability score expressed as the probability of F > 10% in rats [89]. ^c^ Solubility of the molecule in water at 25 °C. ^d^ Percentage of the molecules that would be absorbed through the human intestine [90]. ^e^ Mutagenic potential of the molecule in bacteria [90]. ^1^ Peptides resistant to in silico intestinal digestion. ^2^ Peptides predicted to have all 5 tested bioactivities.

In our case, six peptides contained phenylalanine (FHPR, FSPR, FIPGL, FARPGL, FGPQGG, and FLPPAL) and three peptides contained proline (PCMIR, PFGNR, and PRPMR) at the N-terminal residue. Moreover, peptides with phenylalanine contained other hydrophobic AAs at the C-terminal, such as leucine (L) and glycine (G), suggesting that the positions of the AAs in the sequence could be a key parameter behind the predicted bioactivities.

In silico proteolysis does not guarantee that the resultant bioactive peptides are safe for use in food and pharmaceutical applications. For this reason, we conducted a physicochemical and pharmacokinetic analysis of the selected 27 peptides (Table 3). The SwissADME evaluation provides information on the likelihood that a molecule can be safely used as an oral drug. Ideally, the molecular mass of a potential drug should be less than 500 Da, its lipophilicity should range from −0.7 to 5.0, and its water solubility should not exceed 6 [42]. As can be observed in Table 3, all dipeptides met these criteria and presented a bioavailability score of 0.55, suggesting that these peptides could potentially be absorbed through the intestinal tract. In addition, the other three peptides (ARF, VPGF, and FIPGL) showed intestinal absorption values higher than 20%. One of them, ARF, showed the highest absorption value (71.49%), although its lipophilicity value was not within the ideal range. FIPGL was the potentially bioactive peptide with the largest sequence, and its intestinal absorption value was higher than 20%, meeting SwissADME of drug-likeness criteria. The rest of the peptides exhibited poor intestinal absorption values, and some were predicted to be incapable of being absorbed through the intestinal tract (Table 3). However, these peptides should not be discarded since they could display bioactive effects locally over the cells and tissues of the gastrointestinal tract without being absorbed into the bloodstream [91]. None of the 27 peptides analyzed showed any AMES toxicity.

### 3.4. Molecular Docking of Nannochloropsis gaditana Potential Bioactive Peptides

Since in silico analysis only predicted the potential bioactive peptides released after simulated gastrointestinal digestion of *N. gaditana* proteins, without analyzing their interactions with biological targets, we decided to evaluate the potential mechanism of action through molecular docking of the ACE, DPPIV, and MPO enzymes as a target-specific analysis for the antihypertensive, antidiabetic, and antioxidant activities. Seven peptides showing an intestinal absorption value higher than 20% (GF, FK, FR, RF, ARF, VPGF, and FIPGL) were selected. The Gibbs free energy (ΔG) quantifies the thermodynamic feasibility of a ligand binding to a target, and it is a central parameter related to binding affinity, where the interaction with the lowest ΔG is typically considered the most stable and biologically relevant position [92]. All the dipeptides and the ARF tripeptide showed high binding affinity with the three enzymes tested. Gibbs free energy values ranged from −10.7 to −7.9 kcal/mol for ACE, from −6.7 to −1.7 kcal/mol for DPPIV, and from −8 to −6.6 kcal/mol for MPO, suggesting that they might display their corresponding potential antioxidant, antidiabetic, and antihypertensive activities through the inhibition of these well-described enzymes at a systemic level. In the case of VPGF and FIPGL, both peptides showed high binding affinities for MPO and ACE. The molecular interactions of FIPGL with ACE and MPO enzymes are shown in Figure 4.

FIPGL peptide demonstrated to be resistant to the action of trypsin and chymotrypsin and potentially capable of exerting multifunctional properties, making it an interesting peptide to be further explored. In the case of MPO (Figure 4A,B), FIPGL showed high MPO binding affinity, with a Gibbs free energy value of −9.6 kcal/mol. The peptide–enzyme interaction would be through conventional hydrogen bonds (I_160_, N_162_), alkyl and Pi-alkyl interactions (I_160_, R_31_, A_35_), carbon hydrogen bonds (D_321_), as well as van de Walls forces (Figure 4B). In the human body, MPO normally interacts with hydrogen peroxide (H_2_O_2_) released by the phagocytic cells when they encounter foreign particles, forming a highly reactive enzyme–substrate complex [93]. The elevated MPO activity has been linked to various pathological conditions, increasing the risk of oxidative stress, and previous antioxidant peptides have been developed to use MPO as a therapeutic target [94]. Therefore, the demonstrated interaction between FIPGL and MPO might influence the catalytic turnover or substrate access of MPO, favoring a decrease in its oxidative capacity. This could be a possible mechanism of action behind FIPGL’s in silico predicted antioxidant activity. Nevertheless, other mechanisms of action such as neutralizing ROS, chelating pro-oxidant metal ions, or modulating endogenous antioxidant defense systems could be also behind the predicted antioxidant activity.

In relation to ACE (Figure 4C,D), FIPGL also showed high binding affinity with a Gibbs free energy value of −8.5 kcal/mol. In Figure 4D, FIPGL aminoacidic interactions with ACE were mainly mediated by hydrogen bonds (L_375_, E_376_, T_301_, T_302_, K_449_, S_298_, and Y_287_), alkyl and Pi-alkyl interactions (V_379_, L_427_, L_433_), and van de Walls forces.

ACE converts angiotensin-I into angiotensin-II, a vasoconstrictor, and its overproduction is one of the main causes of high blood pressure in patients with hypertension. Thus, suitable bioactive food-derived peptides have been studied as inhibitors of this enzyme and natural therapeutic alternatives [95]. The presence and positioning of hydrophobic AA residues such as isoleucine, phenylalanine, or proline would be a crucial structural feature for the ACE inhibitory potential [96]. A closer look regarding the structure of ACE has revealed that the active site consists of a zinc coordination site, several hydrogen bonding sub-sites, a proline binding pocket, and a hydrophobic pocket (with “F” residues) with a preference for aromatic AAs like phenylalanine [97]. Consequently, it is imperative for a potent peptide inhibitor to have a hydrophobic residue with an aromatic side chain, such as phenylalanine and proline in its sequence for effective binding. Therefore, FIPGL could be considered a potentially strong ACE inhibitory peptide. Our results were in tune with previous works carried out with other *Nannochloropsis* species like *N. oculata* that have been reported to generate protein hydrolyzates and peptides with ACE1 inhibitory properties [17,98]. Still, quantitative structure–activity relationship models of FIPGL and ACE should be developed to further explore the mechanisms of action behind the predicted antihypertensive activity. Finally, regarding the DPPIV, both peptides did not show any binding affinity, obtaining positive Gibbs free energy. This result suggests that the predicted antidiabetic activity of these peptides could be related to other pathways, such as enhancement of insulin secretion, mimicking of incretin hormones, and/or inhibition of key enzymes involved in carbohydrate digestion, such as α-amylase and α-glucosidase.

Furthermore, in addition to the bioactivities evaluated in the present work, the results underscore different potential applications of *N. gaditana* proteins across industries. While *N. gaditana* is primarily known for its potential as a source of biofuels, the peptides it produces could also play a role in environmental remediation, as peptides with potential antimicrobial or antioxidant properties could be used in bioremediation processes to clean up pollutants or degrade toxic substances in marine environments [99]. In addition to being valuable ingredients for human-health-promoting products, The antioxidant bioactive peptides of *N. gaditana* could also be considered interesting for the formulation of products in the cosmetics and skincare industry or to enhance the growth and health of fish and other marine organisms, highlighting a potential role in the animal feed and aquaculture industry, too [100]. Overall the multifaced nature of *N. gaditana* biomass demonstrates its importance not only as a sustainable biomass source but also as a valuable component in various industries aimed at improving health, sustainability, and environmental protection.

## 4. Conclusions

Our proteomic analysis was able to deepen the scarce knowledge available on the *N. gaditana* proteome. A total of 1955 proteins were identified, of which 1258 were identified with two or more unique peptides. The functional analysis of identified proteins demonstrated that most of them were related to cellular anatomic structures and cellular and binding processes. Furthermore, in silico gastrointestinal digestion and bioactivity prediction revealed the release of small-sized peptides mainly composed of hydrophobic AAs, like phenylalanine and proline, with multiple potential biological activities. The SwissADME analysis showed that these peptides could exert their predicted bioactivities at both local and/or systemic level. Additionally, the molecular docking of potentially absorbable peptides showed high molecular affinity with MPO, ACE, and DPPIV enzymes. It was especially remarkable in the case of the FIPGL peptide, which showed resistance to the action of digestive enzymes, the capability to be absorbed through the gastrointestinal tract, and multifunctionality. This peptide demonstrated strong affinities for MPO and ACE, potential targets of its predicted biological action. Overall, our proteomic and in silico approach suggests, for the first time, the potential of *N. gaditana* as a sustainable source of bioactive peptides with multiple health benefits and potential applications. Therefore, further in vitro assays are needed to confirm the release of the sequences identified by the in silico and docking approaches and their predicted bioactivities.

## Figures and Tables

**Figure 1 foods-14-00252-f001:**
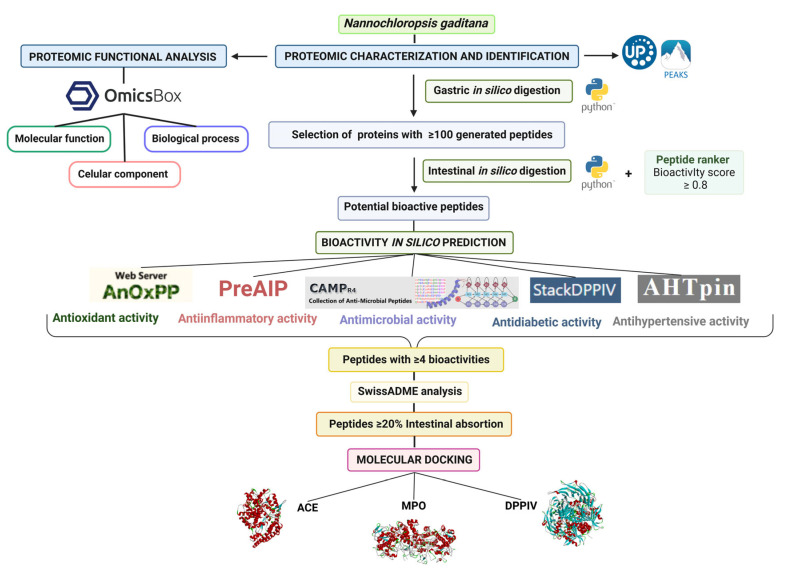
Graphical representation of the workflow followed in the present study.

**Figure 2 foods-14-00252-f002:**
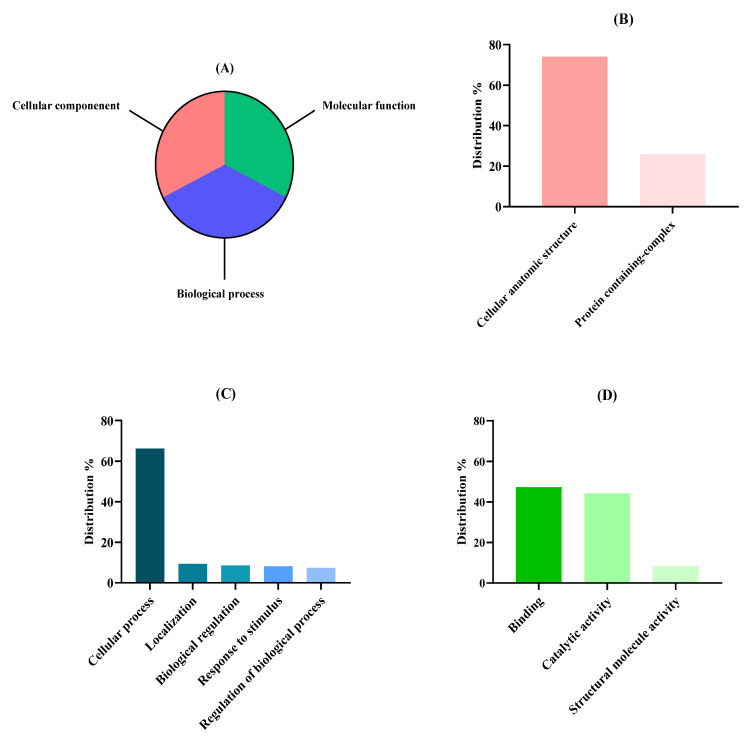
Proteomic functional distribution of identified proteins from *Nannochloropsis gaditana* biomass using gene ontology (GO). (**A**) Proteomic percentage distribution at GO level 1 classification. (**B**) Cellular component GO level 2 distribution. (**C**) Biological process GO level 2 distribution. (**D**) Molecular function GO level 2 distribution.

**Figure 3 foods-14-00252-f003:**
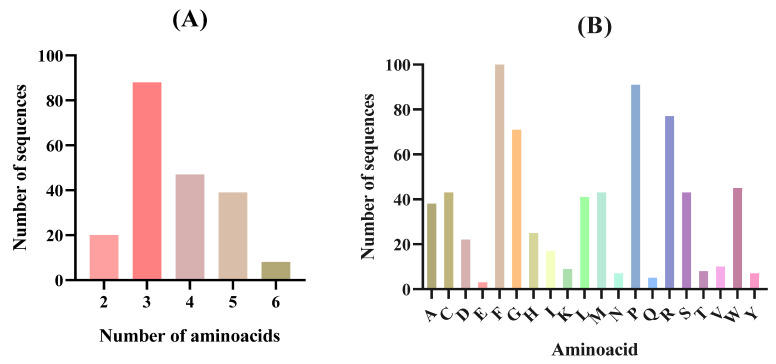
Peptide profile of the peptides released after in silico gastrointestinal digestion from *Nannochloropsis gaditana* biomass and predicted as bioactive using the Peptide Ranker software tool. (**A**) Size distribution expressed as number of amino acids; (**B**) amino acid distribution.

**Figure 4 foods-14-00252-f004:**
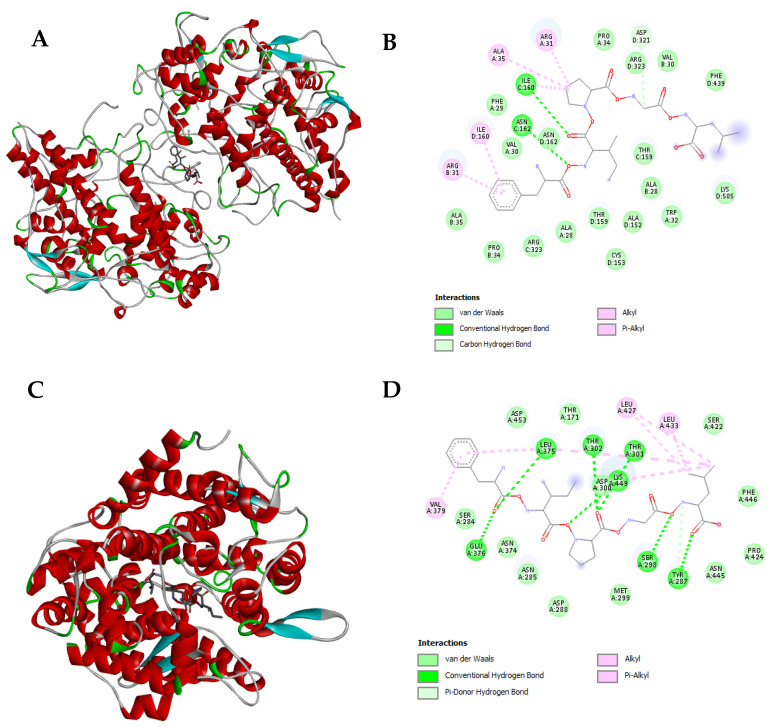
Molecular docking of the potential bioactive peptide FIPGL from *Nannochloropsis gaditana* digest with the human myeloperoxidase (MPO) and the angiotensin converting enzyme (ACE). (**A**) 3D molecular docking representation of FIPGL with MPO; Red: alpha-helices; Blue: beta-strands; Green: loops connecting alpha-helices and beta-strands; Grey: protein backbone or coil regions that do not form distinct secondary structural elements (**B**) interaction type between FIPGL and MPO. Bright green dotted lines: conventional hydrogen bonds; Pink dotted lines: alkyl and Pi-alkyl bonds (**C**) 3D molecular docking representation of FIPGL with ACE; Red: alpha-helices; Blue: beta-strands; Green: loops connecting alpha-helices and beta-strands; Grey: protein backbone or coil regions that do not form distinct secondary structural elements (**D**) interaction type between FIPGL and ACE. Bright green dots lines: conventional hydrogen bonds; Light green dotted lines: Pi-donor hydrogen bonds; Pink dotted lines: alkyl and Pi-alkyl bonds.

**Table 1 foods-14-00252-t001:** Amino acid profile (g/100 g protein and g/100 g biomass) of *Nannochloropsis gaditana*.

Amino Acid	Content		FAO Recommendation (g/100 g Protein)
	g/100 g Protein	g/100 g Biomass	
Essential			
Lysine (K)	4.53 ± 0.09	2.01 ± 0.04	5.20
Tryptophan (W)	n.d.	n.d.	0.70
Phenylalanine (F)	3.62 ± 0.17	1.61 ± 0.08	4.60 ^a^
Tyrosine (Y)	2.42 ± 0.07	1.07 ± 0.03	
Methionine (M)	1.57 ± 0.01	0.70 ± 0.01	2.60 ^b^
Cysteine (C)	0.87 ± 0.12	0.38 ± 0.05	
Threonine (T)	3.40± 0.15	1.51 ± 0.06	2.70
Leucine (L)	5.97 ± 0.01	2.65 ± 0.00	6.30
Isoleucine (I)	2.53 ± 0.03	1.12 ± 0.01	3.10
Valine (V)	3.40 ± 0.02	1.51 ± 001	4.20
Non-essential			
Aspartic acid + Asparragine (D + N)	6.44 ± 0.00	2.85 ± 0.00	
Glutamic acid + Glutamine (E + Q)	8.81 ± 0.09	3.91 ± 0.04	
Serine (S)	3.29 ± 0.15	1.46 ± 0.07	
Histidine (H)	1.38 ± 0.01	0.61 ± 0.00	
Arginine (R)	4.17 ± 0.00	1.85 ± 0.00	
Alanine (A)	5.02 ± 0.01	2.22 ± 0.01	
Proline (P)	6.44 ± 0.01	2.85 ± 0.00	
Glycine (G)	3.55 ± 0.02	1.57 ± 0.01	
EAA	28.31	14.13	
NEAA	39.01	17.28	
TAA	67.41	31.41	
EAA × 100/TAA (%)	42.00		
EAA × 100/NEAA (%)	72.57		
HAA × 100/TAA (%)	47.23		
AAA × 100/TAA (%)	8.96		

n.d.: not detected; ^a^: F + Y; ^b^: M + C; EAA: essential amino acid; NEAA: non-essential amino acid; TAA: total amino acid; HAA: hydrophobic amino acid (A + V + I + L + Y + F + W + M + P + C); AAA: aromatic amino acid (F + W + Y).

**Table 2 foods-14-00252-t002:** *Nannochloropsis gaditana* proteins that generated more than 100 peptides/protein after in silico digestion with pepsin.

Accession ^a^	−10logP ^b^	Description ^c^	Average Mass (KDa)	Peptides Generated After In Silico Gastric Digestion
I2CQP5	426.87	Acetyl-CoA carboxylase	235,196	207
W7U8G3	380.26	ATP-citrate synthase	120,317	102
W7TN63	348.14	Choline dehydrogenase	138,839	113
K8YSL1	287.43	Aminopeptidase N	137,581	102
W7TQD1	264.44	Pyruvate dehydrogenase E1 component subunit alpha	120,314	103
W7TTR4	192.91	P-type atpase	129,122	103
W7UC18	192.19	Phosphoribosylformylglycinamidine synthase	145,158	113
W7TNH0	182.55	Pyruvate carboxilase	134,055	102
K8Z9A0	174.11	Uncharacterized protein	123,187	107
W7TRK7	118.17	Coatomer subunit alpha	141,192	119
W7UBG5	106.82	Ubiquitin-activating enzyme e1	136,993	111
W7U8K2	105.69	Clathrin heavy chain	196,113	163
W7TM9	93.66	Pentatricopeptide repeat containing protein	171,849	132
W7UBN0	87.72	WD40-repeat-containing protein	138,346	110
W7U2D5	80.11	Peptidase M16	143,228	109
W7U0Z1	79.24	Hydantoin utilization protein	146,652	128
W7U5E4	65.46	Carbamoyl-phosphate synthase	168,186	140
W7U2B9	62.39	Zinc finger. ZZ-type	552,820	376
K8YRI3	61.47	DUF2428 domain-containing protein (Fragment)	124,306	102
W7TVB1	56.86	Bromodomain containing 1	243,436	169
W7TYI7	46.56	Cytochrome p450	118,277	117
W7U1T9	46.41	Nuclear receptor corepressor 1-like protein	164,110	122
W7U7L8	45.19	Protease-associated domain PA	142,381	101
W7TPR4	44.49	Tubulin-specific chaperone d	147,094	109
TOTAL				3160

^a^ Accession number of the protein as obtained from the FASTA database. ^b^ PEAKS protein score calculated as the weighted sum of the −10log P scores of the proteins supporting peptides. ^c^ Description of proteins as obtained from the FASTA database.

## Data Availability

The original contributions presented in this study are included in the article/Appendix A. Further inquiries can be directed to the corresponding authors.

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
