# Peer review of "Microalga Nannochloropsis gaditana as a Sustainable Source of Bioactive Peptides: A Proteomic and In Silico Approach"

_foods, 2025, doi:10.3390/foods14020252_

Round 1
Reviewer 1 Report
Comments and Suggestions for Authors
The manuscript by Paterson et al. describes the first in silico analysis in literature of the putative bioactive peptides released from a simulated enzymatic digestion of N. gaditana proteins. The article is well-written and presents a compelling case for future in vitro studies of the microalgae. As a promising alternative to meat-derived protein, the investigation of N. gaditana as a source of health-promoting bioactive peptides is appropriate for the journal Foods. Specific comments follow.
Major Comments
· Many organisms produce bioactive peptides in cellulo, prior to any GI digestion. Often these peptides are impervious to enzymatic digestion in their mature form. The authors have selected solely for full length proteins in this study and may be missing an essential factor in the bioactive potential of the N. gaditana. Please address this in the text or return to the data to examine the possibility of pre-existing small bioactive peptides.
Minor Comments
· In section 2.1, can the authors describe the conditions in which the N. gaditana was grown/extracted at the manufacturer? Would the source material growing conditions represent traditional growth conditions for which a food-source microalgae be grown? (i.e., Is the protein profile of the source material representative of what a food-source N. gaditana would provide?)
· On line 189, the authors sate “…peptides released from proteins that generated ≥ 100 peptides/protein after the gastric phase were selected to be concurrently digested with trypsin…” Can the authors clarify why a threshold of >/= 100 peptides/protein was chosen as a metric for further in silico digestion?
· On line 400, the authors state that most of the peptides identified in Table 3 have not been previously reported in literature. Do these sequences appear in other organisms with potential or current food value?
Reviewer 2 Report
Comments and Suggestions for Authors
the present study entitled "Microalga Nannochloropsis gaditana as a sustainable source of bioactive peptides: a proteomic and in silico approach" has practical significance. the study is well explained and organized. fewer suggestions can further improve its
line 93-95: in vitro experiments are not expensive as author stated. science can not rely only on docking experiments.
section 2.1: author should describe the details of microalgae cultivation conditions.
authors should strengthen the discussion section with more potential application of this microalgae.
conclusion should be supported by limitation of the study.
line 237: author should describe the reason of predominancy of glutamic acid.
